# Reconstructing Humans with Articulated Hands using Transformers

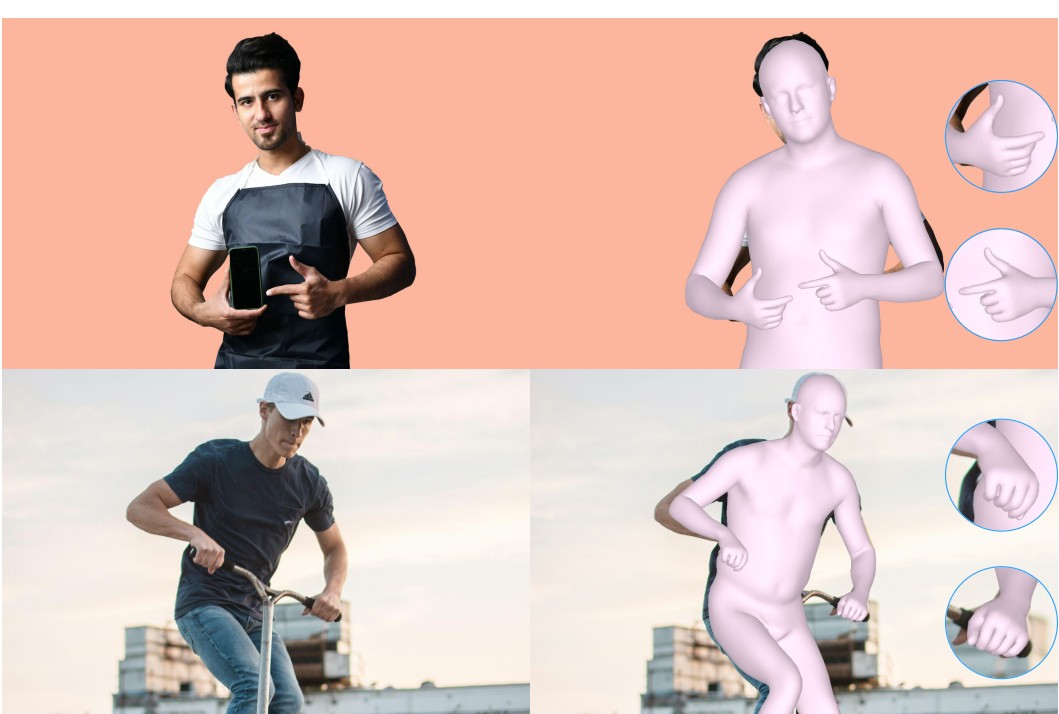

Figure 1: **Reconstructing expressive humans with articulated hands in 3D**. We propose BodhaHMR, an approach for **Bod**y and **Ha**nd **H**uman **M**esh **R**ecovery from a single input image. BodhaHMR delivers consistent estimates of body and hand pose by leveraging two dedicated backbones, one for body processing and one for hand processing. The features produced from each backbone are fused together by a unifying transformer decoder to predict the body and hand pose parameters in the form of the SMPL-H model (Romero et al., 2017).

## ABSTRACT

In this paper, we introduce an approach to reconstruct 3D humans with expressive hands given a single image as input. Current methods for pose estimation display robust performance for either bodies or hands. Unfortunately, these methods fail to simultaneously produce accurate 3D body and hand reconstructions. To address this limitation, we take a more cohesive approach to ensure both coarser and finer features of the human body are properly localized. Our approach is based on a feedforward network and following recent best practices, we adopt a fully transformer-based architecture. One of the key design choices we make is to leverage two separate backbone networks, one for 3D human pose and one for 3D hand pose estimation. These backbones process independently the body region and the hand regions and can make estimates about the bodies and the hands of the person. However, when the estimates are made independently, they tend to be inconsistent with one another and lead to unsatisfying reconstruction. Instead, we introduce a coupling transformer decoder that is trained to consolidate the intermediate features from the individual backbones into making a consistent estimate

for the body and the hands. The full system is trained on multiple datasets, including images with body ground truth, with hand ground truth, as well as images that include both body and hand ground truth. We evaluate our approach on the AGORA, ARCTIC, and COCO datasets, reporting metrics for both bodies and hands reconstruction accuracy to highlight our model's robustness over previous baselines.

# 1 INTRODUCTION

The reconstruction of expressive 3D human pose and shape from a monocular image requires a consistent understanding of various features of the presented human body, both coarse (e.g. head, torso, arms, legs) and fine (e.g. hands, fingers, feet, toes, eyes). Accordingly, a robust estimation method should appropriately respect features at both levels. When this is not the case, we observe clear discrepancies in the generated meshes. For instance, a system may implicitly prioritize fitting the predicted head and in turn sacrifice the prediction's hand pose and alignment, or vice versa. As human pose and shape estimation has numerous applications in robotics, healthcare, AR/VR, and sports, these inconsistencies can have noticeable consequences for downstream applications.

Breakthroughs in 3D pose estimation have recently relied on the robustness provided by coupling large models with large-scale data (Xu et al., 2022b; Goel et al., 2023; Pavlakos et al., 2024). This has also enabled significant progress in methods for 3D body (Goel et al., 2023; Dwivedi et al., 2024; Fiche et al., 2025; Su et al., 2025; Zhang et al., 2025) and 3D hand reconstruction (Pavlakos et al., 2024; Potamias et al., 2024; Chen et al., 2025; Fan et al., 2025). However, when it comes to jointly reconstructing humans with articulated hands, many recent approaches (Baradel* et al., 2024; Yin et al., 2025) tend to underperform. Specifically, they typically place less emphasis on finer features to better align coarser aspects of the body. Our aim is to address these limitations and make a more balanced and consistent 3D pose prediction by carefully considering coarse and finer features.

We present BodhaHMR, a method that takes crops of a person's body and hands from a single image as input, and performs **Bod**y and **Ha**nd **H**uman **M**esh **R**ecovery. In particular, our approach adopts two backbones: one for bodies and one for hands. These backbones take the corresponding crops as input and estimate features for the body and the two hands. The backbones are initialized using weights from a state-of-the-art approach for 3D body reconstruction (Goel et al., 2023) and a state-of-the-art approach for 3D hand reconstruction (Pavlakos et al., 2024). BodhaHMR then uses a coupling transformer decoder that takes as input the body and hand features and regresses the SMPL-H (Romero et al., 2017) parameters. As shown in Figure 1, BodhaHMR makes a unified and consistent estimate of a person's body and hands. Furthermore, BodhaHMR is powered by a large model and large data. BodhaHMR adopts a transformer-based architecture (Dosovitskiy et al., 2021; Xu et al., 2022b), while our training data consists of in-the-wild (Jin et al., 2020; Xu et al., 2022a), synthetic (Hewitt et al., 2024), and controlled (Zhu et al., 2023; Ionescu et al., 2014; 2011) datasets. 3D ground truth for bodies and hands is available in the synthetic and controlled data, while the in-the-wild data has 2D ground truth for hands and bodies. Together, these design decisions comprise a comprehensive approach for expressive mesh reconstruction.

We evaluate our method across a variety of datasets to demonstrate its robustness. For 3D pose (3D body and hand keypoints), we benchmark our approach on the ARCTIC (Fan et al., 2023) and AGORA (Patel et al., 2021) datasets. ARCTIC provides many challenging and expressive hand poses in a studio setting, while AGORA introduces synthetic subjects and environments, with some being low-resolution. For 2D pose, in addition to ARCTIC and AGORA, we benchmark on images from COCO-Wholebody (Jin et al., 2020; Xu et al., 2022a). COCO offers many diverse in-the-wild examples for our testing.

We contribute BodhaHMR, a cohesive and holistic approach for expressive human mesh recovery from a single RGB image. Moreover, we perform extensive evaluation on BodhaHMR and alternative state-of-the-art approaches. Our testing reveals that BodhaHMR outperforms state of the art across the board on 2D hand pose while maintaining state-of-the-art performance on bodies.

## 2 RELATED WORK

**3D human body pose and shape estimation.** Human mesh recovery has been considered from various perspectives, but we will limit our scope to approaches that derive pose and shape parameters from a single image. Previous optimization-based approaches estimate 3D human pose and shape by analyzing 2D image features (Guan et al., 2009; Bogo et al., 2016; Lassner et al., 2017; Omran et al., 2018; Pavlakos et al., 2019a; Xu et al., 2020). Since the introduction of HMR (Kanazawa et al., 2018), which learned to estimate SMPL parameters using a convolutional neural network (CNN), regression-based approaches have become more prevalent (Pavlakos et al., 2019b; Guler & Kokkinos, 2019; Jiang et al., 2020; Georgakis et al., 2020; Kolotouros et al., 2021; Kocabas et al., 2021; Zhang et al., 2021).

Recently, there has been a shift from CNNs to transformer-based (Vaswani et al., 2017) architectures. Mesh Graphformer (Lin et al., 2021) directly estimates the mesh vertices using transformers. HMR 2.0 (Goel et al., 2023), on the contrary, regresses SMPL parameters without domain-specific design decisions to outperform previous baselines. TokenHMR (Dwivedi et al., 2024), VQ-HPS (Fiche et al., 2024), and MEGA (Fiche et al., 2025) use a Vector Quantized-Variational Autoencoder (VQ-VAE) to tokenize human pose related information and minimize irregular predictions. ADHMR (Shen et al., 2025) iterates on existing diffusion-based HMR methods by performing direct preference optimization (Wallace et al., 2023). DeforHMR (Heo et al., 2025) uses a novel query-agnostic implementation of deformable attention transformers (Xia et al., 2023; 2022; Zhu et al., 2020) to enhance the model's spatial awareness. PAMA (Chen & He, 2025) couples a module for limb appearance consistency with full-perspective projection and an adapted reprojection loss to handle alignment discrepancies. Other approaches focus on optimizing for different challenges of human pose estimation. For instance, MetricHMR (Zhang et al., 2025) regresses 3D position information in addition to human pose and shape parameters, using a camera ray representation method. SAT-HMR (Su et al., 2025) tackles the issue of high computational costs in one-stage multi-person pose estimation. Our approach is different to these approaches by aiming to reconstruct not only the body, but also the hands of the person.

**3D hand pose and shape estimation.** Similarly to the above, we will focus on approaches that recover hand pose and shape parameters from a single image. Most of the initial methods (Baek et al., 2019; Zhang et al., 2019; Boukhayma et al., 2019; Park et al., 2022; Oh et al., 2023) for this problem employ the MANO parametric model (Romero et al., 2017) and regress the parameters. Other early approaches estimate the mesh vertices instead of regressing a parametric model (Ge et al., 2019; Chen et al., 2022; Jiang et al., 2023). As with body pose estimation, a similar shift to transformer architectures for hand pose estimation has occurred. HaMeR (Pavlakos et al., 2024) adopts a vanilla transformer-based regression network, surpassing previous approaches without a specialized design. Many of these transformer-based approaches make intentional augmentations to resolve critical challenges. WiLoR (Potamias et al., 2024) deploys a coarse-to-fine method by passing predicted parameters through a refinement module, to improve alignment and handle occlusions. To achieve high computational efficiency, simpleHand (Zhou et al., 2024) decomposes its mesh decoder into token generator and mesh regressor modules, and develops a streamlined structure. HHMR (Li et al., 2024b) uses a graph diffusion model in combination with attention mechanisms to construct a flexible and robust framework capable of many mesh recovery tasks. HandOS (Chen et al., 2025) builds a one-stage pipeline to detect hands and estimate pose, without directly regressing the MANO parameters. Fan et al. (2025) utilizes temporal information to improve performance on low-resolution images. To increase estimation accuracy, EHPE (Zheng et al., 2025) focuses on analyzing the distal phalanx tip and wrist joints, while Karvounas et al. (2025) incorporate texture-based supervision into existing estimation methods. Instead of transformers, Hamba (Dong et al., 2024) opts for a Mamba-based (Gu & Dao, 2023; Dao & Gu, 2024) architecture complete with graph learning and state space modeling. Although we use insights from the hand pose estimation literature and we adopt the HaMeR backbone (Pavlakos et al., 2024) in our architecture, we are different to these approaches by estimating a holistic body and hand reconstruction.

**3D expressive human mesh recovery.** The body approaches detailed so far focus on the SMPL parametric model, which does not capture the hand articulation or facial expressions. Earlier attempts at expressive mesh recovery (Choutas et al., 2020; Moon et al., 2022; Zhang et al., 2023) regress the SMPL-X parametric model (Pavlakos et al., 2019a) parameters to capture more nuances

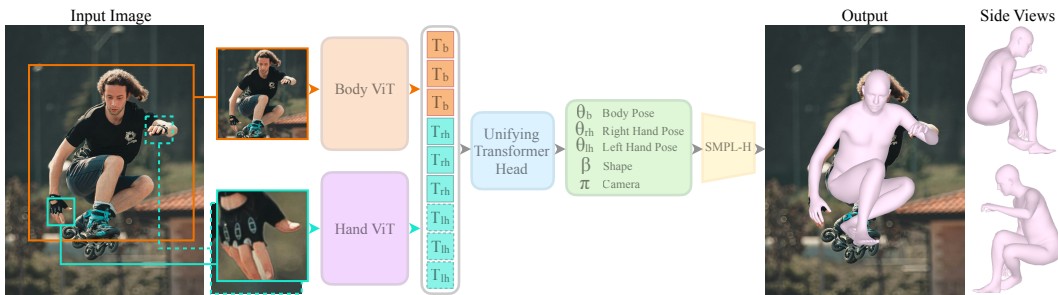

Figure 2: **Overview of BodhaHMR's architecture**. We present a transformer-based method of expressive human mesh recovery. Our network takes crops from an image of a person's body, right hand, and left hand as input, and passes these through two vision transformer backbones. Specifically, the body crop is passed through a dedicated backbone for bodies, while the hand crops are passed through a dedicated backbone for hands. The tokens generated from the body $T_b$, right hand $T_{rh}$, and the left hand $T_{lh}$ are concatenated along the token dimension and passed to a unifying transformer head, which ultimately outputs accurate SMPL-H (Romero et al., 2017) parameters.

in subjects' hands and faces. Recently, we have observed the same transformer shift in this context when it comes to the network architecture. SMPLer-X (Cai et al., 2023) and its follow up, SMPLest-X (Yin et al., 2025), primarily demonstrate the positive impacts of scaling up data and using large Vision Transformers (ViT) (Dosovitskiy et al., 2021; Xu et al., 2022b). OSX (Lin et al., 2023) designs a Component Aware Transformer (CAT) to provide a one-stage framework to retrieve face, hand, and body parameters that ensures good connectivity between subject attributes. Multi-HMR (Baradel* et al., 2024) and AiOS (Sun et al., 2024) present single-shot approaches to deterministically deal with the challenge of multi-person expressive pose estimation while retaining accurate 3D locations. On the other hand, CondiMen (Romain et al., 2025) deploys a probabilistic method to address multi-person pose estimation. PromptHMR (Wang et al., 2025) enables pose and shape estimation through both spatial and semantic prompts to preserve visual context. Moving away from transformers, D-PoSE (Vasilikopoulos et al., 2024) uses a lighter CNN backbone in combination with depth and part-segmentation maps to outperform competing transformer-reliant methods.

Following the trend in body, hand, and expressive pose estimation, we use a transformer architecture to recover body and articulated hand parameters. Unlike the vast majority of techniques for expressive mesh recovery, we choose to regress the SMPL-H parametric model (Romero et al., 2017) instead of SMPL-X (Pavlakos et al., 2019a). At a first glance, this looks like a more constrained setting, but we are motivated by the need for very precise body and hand pose estimation for multiple downstream applications, particularly in robotics (Li et al., 2024a; Fu et al., 2024; Wang et al., 2024).

## 3 TECHNICAL APPROACH

In this section, we present the technical details of our approach. First we provide some preliminaries about the SMPL-H model (Section 3.1), and then we elaborate on our approach (Section 3.2), the architecture we use (Section 3.3) and the training losses (Section 3.4).

### 3.1 SMPL-H MODEL

For our reconstruction, we employ the SMPL-H parametric model (Romero et al., 2017) of the human body and hands. SMPL-H takes as input pose ($\theta \in \mathbb{R}^{52 \times 3 \times 3}$) and shape ($\beta \in \mathbb{R}^{10}$) parameters and outputs a mesh $M \in \mathbb{R}^{3 \times V}$ of the human body with articulated hands, where $V = 6890$ vertices. The pose parameters include body pose parameters $\theta_b \in \mathbb{R}^{21 \times 3 \times 3}$, global orientation $\theta_g \in \mathbb{R}^{3 \times 3}$, and hand pose parameters for the left and the right hand ($\theta_{lh}, \theta_{rh} \in \mathbb{R}^{15 \times 3 \times 3}$). The joints $X \in \mathbb{R}^{3 \times k}$ are a linear combination of the vertices.

## 3.2 Human mesh reconstruction with expressive hands

Our aim is to use an RGB image of a person to generate a representative, articulated 3D mesh. To accomplish this, we learn a predictor function $f(I)$ that maps the image to the SMPL-H parameters $\Theta = (\theta, \beta, \pi)$. This function also produces camera parameters $\pi = (R, t)$ as part of the output parameters. Our camera model has fixed focal length and camera intrinsics $K$. With global orientation $R \in \mathbb{R}^{3 \times 3}$ and translation $t \in \mathbb{R}^3$, we can project the 3D joints $X$ onto the image using $x = \pi(X) = \Pi(K(RX + t))$. Note that $\theta$ contains a global orientation, so we fix $R$ to be the identity in practice. Ultimately, we learn $f(I) = \Theta$.

## 3.3 Architecture

Following Goel et al. (2023) and Pavlakos et al. (2024), our approach utilizes a "transformerized" design. As shown in Figure 2, the architecture comprises two Vision Transformer huge (ViT-H) backbones (Dosovitskiy et al., 2021) (one for bodies, one for hands) and a unifying transformer decoder head. The coupling decoder cohesively considers body and hand features before making a prediction. This significantly contrasts having separate decoders for the body and hands, which make predictions for the bodies and hands in isolation, and require us to fuse the estimates together afterwards. We take three crops of the bounding boxes of the body ($I_b$), right and left hands ($I_{rh}, I_{lh}$) from the original image and transform each one into $16 \times 16$ patches to feed into the corresponding backbones. The ViT-H backbones convert the patches into a total of three sets of tokens, one set per crop. We concatenate these token outputs along the token dimension and feed them all into the decoder, which cross-attends to them. Finally, the unifying transformer head outputs $\Theta$ corresponding to the given subject.

## 3.4 Losses

Our model is trained with a combination of 2D and 3D losses, mirroring best practices from prior literature (Kanazawa et al., 2018; Kolotouros et al., 2019; Goel et al., 2023). The datasets we train with contain a variety of annotations, so we deploy a subset of the losses below for each training example. If ground-truth SMPL-H pose ($\theta^*$) and shape ($\beta^*$) parameters are available, we use the following MSE loss:

$$\mathcal{L}_{smplh} = ||\theta_b - \theta_b^*||_2^2 + ||\theta_g - \theta_g^*||_2^2 + ||\theta_{rh} - \theta_{rh}^*||_2^2 + ||\theta_{lh} - \theta_{lh}^*||_2^2 + ||\beta - \beta^*||_2^2. \quad (1)$$

With this loss formulation, we can straightforwardly handle partial annotations by "turning off" missing terms. If ground-truth 3D joint annotations $X^*$ are available, we use the following L1 loss on the 3D keypoints:

$$\mathcal{L}_{kp3D} = ||X - X^*||_1. \quad (2)$$

If ground-truth 2D keypoint annotations $x^*$ are available, we use the following reprojection L1 loss on the projected 3D keypoints:

$$\mathcal{L}_{kp2D} = ||x - x^*||_1. \quad (3)$$

Lastly, it is possible that the model predicts abnormal 3D poses when ground-truth pose or joint annotations are not available. We combat this by training discriminators $D_k$ for the (i) body pose parameters $\theta_b$, (ii) shape parameters $\beta$, and (iii) each individual body joint angle, similar to Kanazawa et al. (2018). The corresponding generator loss is:

$$\mathcal{L}_{adv} = \sum_k (D_k(\theta_b, \beta) - 1)^2. \quad (4)$$

# 4 Experiments

## 4.1 Preliminaries

**Datasets and implementation.** We train our BodhaHMR on COCO-Wholebody (Jin et al., 2020; Xu et al., 2022a), Human3.6M 3D WholeBody (Zhu et al., 2023; Ionescu et al., 2014; 2011), and SynthMoCap (SynthBody and SynthHand) (Hewitt et al., 2024). These datasets offer a blend of in-the-wild, controlled, and synthetic examples to generate a robust final model. We utilize the pretrained weights of HMR 2.0b (Goel et al., 2023) and HaMeR (Pavlakos et al., 2024) for body

| Method | AGORA | | | | ARCTIC | | | | COCO | |
|---|---|---|---|---|---|---|---|---|---|---|
| | MPJPE ↓ | PA-MPJPE↓ | @0.05↑ | @0.1↑ | MPJPE ↓ | PA-MPJPE↓ | @0.05↑ | @0.1↑ | @0.05↑ | @0.1↑ |
| Frankenstein (Hu, 2025) | 58.4 | 11.8 | 0.050 | 0.171 | 25.8 | 10.1 | 0.112 | 0.351 | 0.069 | 0.206 |
| SMPLest-X (Yin et al., 2025) | 51.0 | 10.0 | 0.012 | 0.047 | 41.9 | 15.7 | 0.013 | 0.052 | 0.012 | 0.047 |
| Multi-HMR 896L* (Baradel* et al., 2024) | - | - | - | - | 35.1 | 13.6 | 0.038 | 0.137 | 0.013 | 0.049 |
| Multi-HMR 672L* (Baradel* et al., 2024) | - | - | - | - | 40.3 | 13.0 | 0.036 | 0.126 | 0.011 | 0.042 |
| Ours | 47.2 | 10.8 | 0.105 | 0.312 | 30.2 | 11.3 | 0.121 | 0.365 | 0.120 | 0.361 |

Table 1: **Comparison with the state-of-the-art on hand reconstruction.** We evaluate the predicted 3D and reprojected 2D hand joints by taking the average of metrics from both hands. We report MPJPE, PA-MPJPE, PCK @0.05, and PCK @0.1 if 3D hand ground truth is available (AGORA and ARCTIC). If only 2D hand ground truth annotations are available (COCO), PCK @0.05 and PCK @0.1. Our method outperforms most state of the art in the 3D evaluation. The Frankenstein approach produces more accurate 3D hands in some cases because it copies hands from HaMeR (Pavlakos et al., 2024). BodhaHMR achieves the best 2D results across all datasets. * denotes methods trained on the AGORA training split. All MPJPE and PA-MPJPE metrics are in mm.

| Method | AGORA | | | | ARCTIC | | | | COCO | |
|---|---|---|---|---|---|---|---|---|---|---|
| | MPJPE ↓ | PA-MPJPE↓ | @0.05↑ | @0.1↑ | MPJPE ↓ | PA-MPJPE↓ | @0.05↑ | @0.1↑ | @0.05↑ | @0.1↑ |
| Frankenstein (Hu, 2025) | 157.9 | 67.0 | 0.801 | 0.915 | 163.5 | 77.3 | 0.762 | 0.904 | 0.806 | 0.941 |
| SMPLest-X (Yin et al., 2025) | 101.4 | 61.1 | 0.657 | 0.855 | 95.6 | 41.2 | 0.761 | 0.948 | 0.548 | 0.831 |
| Multi-HMR 896L* (Baradel* et al., 2024) | - | - | - | - | 125.0 | 56.6 | 0.679 | 0.856 | 0.530 | 0.772 |
| Multi-HMR 672L* (Baradel* et al., 2024) | - | - | - | - | 115.9 | 53.4 | 0.679 | 0.881 | 0.473 | 0.713 |
| Ours | 142.0 | 61.6 | 0.765 | 0.909 | 108.3 | 51.5 | 0.755 | 0.949 | 0.718 | 0.915 |

Table 2: **Comparison with the state-of-the-art on body reconstruction.** We report results on body reconstruction accuracy. Our method maintains consistent state-of-the-art performance on 3D and 2D evaluation. * indicates methods trained on the AGORA training split. All MPJPE and PA-MPJPE numbers are in mm.

and hands backbones, respectively. These backbones provide good initialization for their respective tasks, as they have been trained on numerous body and hand examples. During training, we freeze these backbones and only train the unifying transformer head parameters. To encourage better alignment of the mesh hands to the image, we take inspiration from Pavlakos et al. (2019a) and Choutas et al. (2020), and gradually increase the 2D hand loss weights.

**Baselines.** For our evaluation, we compare against state-of-the-art methods for expressive mesh recovery. We examine the performance of two ViT-L (Dosovitskiy et al., 2021; Xu et al., 2022b) based approaches from Multi-HMR (Baradel* et al., 2024), which process input images at resolutions of 896x896 (896L) and 672x672 (672L), respectively. Furthermore, we report results from the publicly available checkpoint of SMPLest-X (Yin et al., 2025). Lastly, to compare with a straightforward approach of fusing the results from an independent network for bodies and an independent network for hands, we design another "Frankenstein" baseline. For each example, we use HMR 2.0 to estimate $\theta_b, \theta_g, \beta$ and HaMeR to estimate $\theta_{rh}, \theta_{lh}$. Straightforwardly generating a SMPL-H mesh with these parameters results in poor hand alignments due to discrepancies between the estimated hand wrist joints and the estimated body wrist joints. To achieve more consistent body and hand integration, we recompute the wrist poses by taking the elbow rotations from HMR 2.0, the hand rotations from the HaMeR, and applying the logic from Hu (2025).

**Metrics.** We employ the AGORA (Patel et al., 2021) and ARCTIC (Fan et al., 2023) allocentric validation sets to evaluate 3D pose prediction. In particular, we report Mean Per Joint Position Error before (MPJPE) (Ionescu et al., 2014) and after Procrustes Alignment (PA-MPJPE) (Kanazawa et al., 2018; Zhou et al., 2019). To evaluate 2D pose accuracy, we use the ARCTIC allocentric, AGORA, and COCO-Wholebody validation sets. We report the Percentage of Correct Keypoints (PCK) (Yang & Ramanan, 2013) metric at thresholds of 0.05 and 0.1 for the body and hands.

## 4.2 HAND RESULTS

We report the results of hand reconstruction in Table 1. In our evaluation, BodhaHMR model excels in the context of hand reconstruction. Our approach outperforms the other baselines on all given metrics, except for the 3D hand pose evaluation on ARCTIC and PA-MPJPE on AGORA. In these cases, we see that BodhaHMR does better than its competitors on the 2D pose evaluation. As the

| Models | AGORA | | | | ARCTIC | | | | COCO | |
|---|---|---|---|---|---|---|---|---|---|---|
| | MPJPE ↓ | PA-MPJPE↓ | @0.05↑ | @0.1↑ | MPJPE ↓ | PA-MPJPE↓ | @0.05↑ | @0.1↑ | @0.05↑ | @0.1↑ |
| Single Backbone | 53.7 | 11.2 | 0.0283 | 0.106 | 47.5 | 17.7 | 0.0426 | 0.158 | 0.0416 | 0.156 |
| BodhaHMR | **51.1** | **10.3** | **0.0589** | **0.201** | **40.3** | **12.3** | **0.0528** | **0.185** | **0.0613** | **0.214** |

Table 3: **Ablation on hand backbone.** We study the impact of introducing the hand backbone into the network. The single backbone approach freezes the pretrained weights from Goel et al. (2023) during training and regresses the SMPL-H parameters from one crop of the subject. We compare this against BodhaHMR on hand reconstruction. The single backbone version performs consistently worse than our method on these benchmarks, highlighting the importance of the hand backbone.

2D pose evaluation measures the reprojection of the 3D joints on to the image, BodhaHMR's hand localization on the image is more accurate than Frankenstein and SMPLest-X, respectively. This demonstrates the effectiveness of BodhaHMR's consolidated approach: our method consistently estimates more accurate 2D hand positioning than competitors. Moreover, for the 2D hand metrics, we observe that Frankenstein is consistently the second-best approach. Our method realizes gains of 1.5× to 2× over Frankenstein for AGORA and COCO. On ARCTIC, BodhaHMR outperforms Frankenstein by 3% to 7%. The in-the-wild images from the COCO dataset have more subjects, interactions, environments, and perspectives than the studio images from ARCTIC. AGORA also presents images with varied and numerous subjects in a variety of settings. In contrast, ARCTIC's validation split only contains one subject in a controlled, studio setting. Therefore, we believe 2D performance on COCO and AGORA provides a better indication of a model's generalizability and robustness than ARCTIC.

### 4.3 BODY RESULTS

We report the results of body reconstruction in Table 2. As we see, our approach lags behind only SMPLest-X, and beats out the other methods in 3D body pose. Note that the body joints of the SMPL-H and SMPL-X parametric models are localized differently, and AGORA and ARCTIC provide ground-truth annotations with respect to the SMPL-X model. Despite this discrepancy, BodhaHMR's performance on the 3D body evaluation competes with state-of-the-art methods that regress SMPL-X parameters, demonstrating the robustness of our model. Similarly, on AGORA and COCO 2D evaluation, BodhaHMR outperforms SMPLest-X and Multi-HMR and slightly falls behind the Frankenstein method. Given these results, BodhaHMR maintains state-of-the-art performance on the body metrics. This is significant when contextualized with the hand results above: BodhaHMR produces the most consistent and balanced predictions across all tested methods.

### 4.4 ABLATION STUDY

We investigate the impact of utilizing distinct backbones for hands and bodies on the hand accuracy in Table 3. To examine this, we compare against a model that adopts the encoder backbone from Goel et al. (2023) and freezes these weights while training. At test time, it passes a crop of a person's body through the backbone, which produces one set of image tokens as output. These tokens are passed through a standard decoder head, which cross-attends to them and regresses the SMPL-H parameters. As we see, our method outperforms the straightforward extension of HMR 2.0 across the board: BodhaHMR provides more accurate 3D hand pose reconstructions and better 2D alignments. Our method is superior in estimating both the hand pose and 2D hand location than the simpler model, justifying the use second backbone for the hands. Please refer to Section A.4 for more implementation details and body results of this ablation.

### 4.5 QUALITATIVE RESULTS

We perform a qualitative comparison of our approach with state-of-the-art in Figure 3. These results support our findings from the quantitative evaluation, as BodhaHMR demonstrates more accurate 2D hand location in combination with similar body estimates as the other methods. In fact, due to BodhaHMR's unified approach, it achieves significantly better image alignment in the egocentric example. In Figures 4 and 5, we provide more qualitative results from BodhaHMR. Our method produces well-aligned bodies and hands under various viewpoints, occlusions, and environments.

| Input Image | Multi-HMR | SMPLest-X | Frankenstein | Ours |
|---|---|---|---|---|

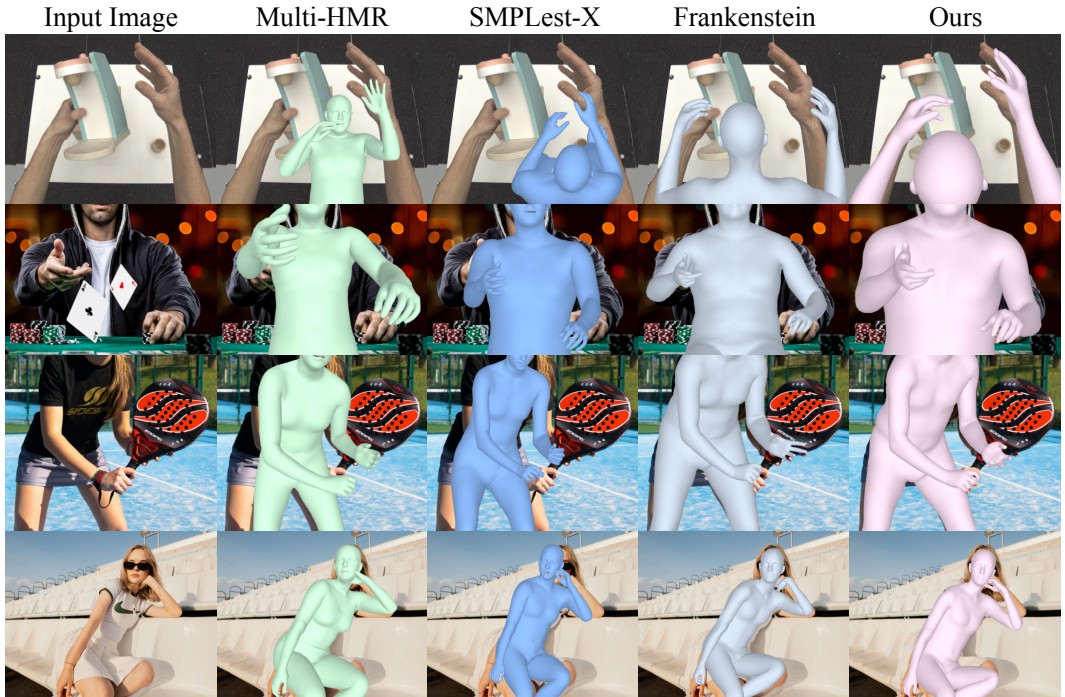

Figure 3: **Qualitative comparison with state-of-the-art**. We show results from the 896L version of Multi-HMR. In challenging egocentric and in-the-wild settings, our model recovers more accurate hand alignments and articulations than competitors. Please zoom in for details.

| Input Image | Recovered Mesh | Side View | Input Image | Recovered Mesh | Side View |
|---|---|---|---|---|---|

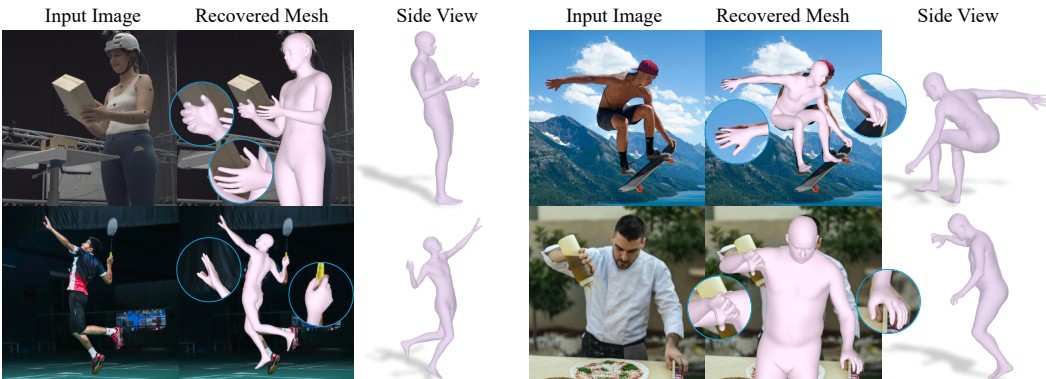

Figure 4: **Novel views of BodhaHMR**. For each input image, we show the recovered mesh, hand close-ups, and a side view. Our method gracefully handles a variety of poses and occlusions.

## 5    CONCLUSION

We describe BodhaHMR, a method for reconstructing 3D expressive humans from a single RGB image. Our approach builds on state-of-the-art methods for bodies and hands: BodhaHMR consolidates the body and hand features from these networks to achieve notable improvements on 2D hand alignment, without sacrificing state-of-the-art performance on the body.

**Limitations.**    One limitation of BodhaHMR is we notice some of the reconstructed hands could be more expressive and located even more precisely. This may be the result of the unifying decoder incorrectly localizing the tokens from the hand backbones on the input image. Future work could provide more context to the unifying transformer head using positional embeddings (Prakash et al., 2024) to potentially mitigate these issues. Additionally, while we provide some qualitative egocen-

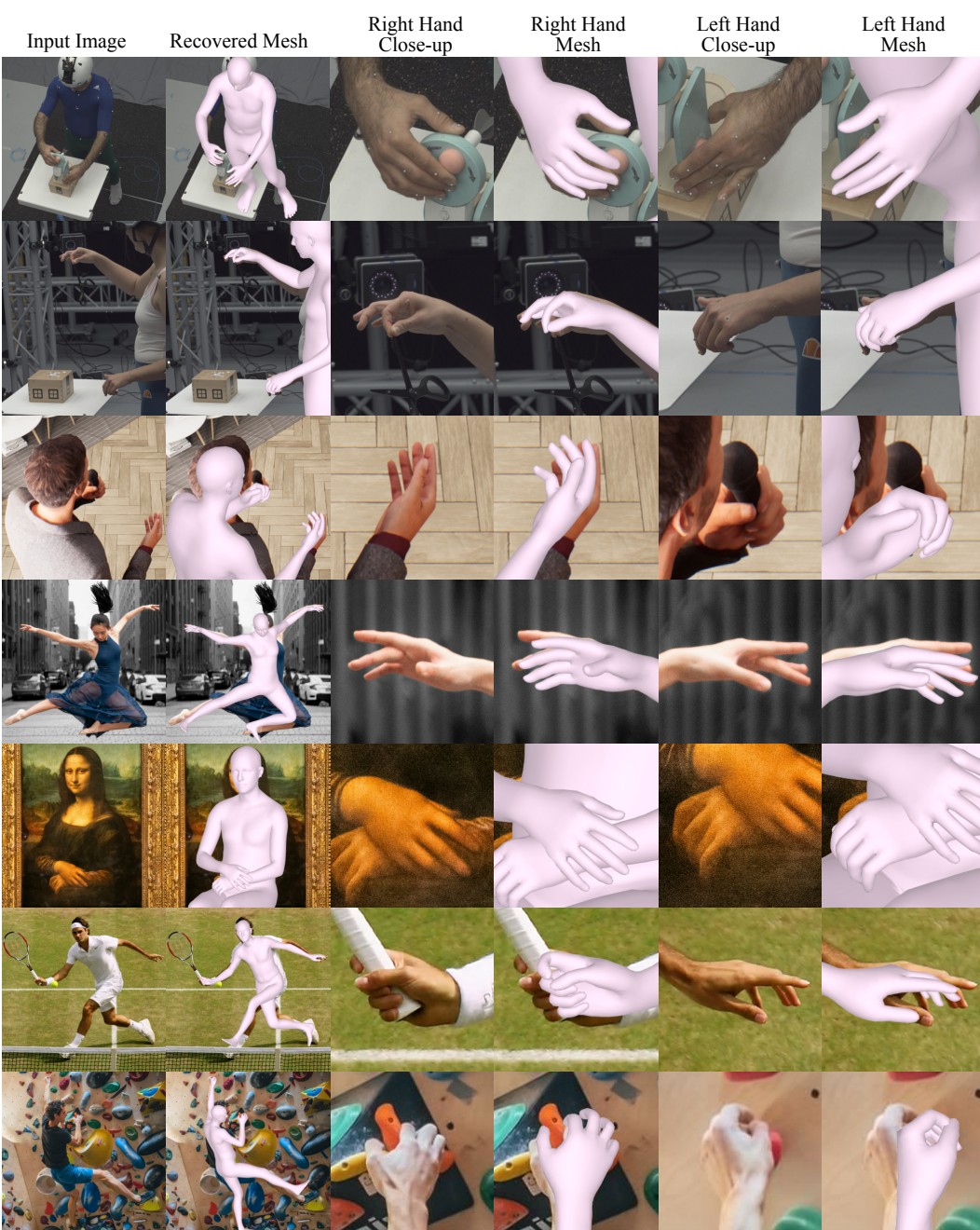

Figure 5: **Qualitative results of BodhaHMR**. For each input image, we display the overlaid reconstruction, close-ups of the input hands, and their reconstructions. Rows 1-2 are from ARCTIC, row 3 is from AGORA, and rows 4-7 are from the Internet. Our approach recovers cohesive and expressive hands alongside robust bodies.

tric examples with positive results, our method could perform even more robustly in this context. We observe good performance due to the synthetic hand training data, which contains isolated hand examples that effectively mimic egocentric images. Future work could explore explicitly adding more egocentric examples and datasets to improve performance in this context.

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

# A APPENDIX

## A.1 ARCHITECTURE

As detailed in the main manuscript, BodhaHMR uses two ViT-H (Dosovitskiy et al., 2021) image encoders. We take the backbone for bodies from HMR 2.0 (Goel et al., 2023) and the backbone for hands from HaMeR (Pavlakos et al., 2024). Both backbones accept input images of size $256 \times 192$ and output $16 \times 12$ image tokens, each of dimension 1280. At test time, we pass a crop of the person's body to the body backbone and the crops of their hands sequentially to the hands backbone. The three sets of tokens produced by the encoders are concatenated along the token dimension and passed to the coupling transformer decoder. For the unifying decoder, we adopt the same architecture as Goel et al. (2023). It has 6 layers and takes a 1024-dimensional SMPL-H query token as input and cross-attends to the concatenated image tokens. The unifying decoder has 8 (64- dim) heads for cross-attention and self-attention and a hidden dimension of 1024. We readout $\theta_b, \theta_g, \theta_{rh}, \theta_{lh} \beta$, and $\pi$ from the output token of the coupling decoder.

## A.2 TRAINING

During training, we use a weight of 0.13 to sample examples from Human3.6M, 0.12 for Human3.6M with hands, 0.25 for SynthBody, 0.25 for SynthHand, 0.13 for COCO, and 0.12 for COCO with hands. Note that we have two versions of the COCO and Human3.6M datasets: one version containing only body annotations and one version with body and hand annotations. Moreover, we train our final model with AdamW (Kingma & Ba, 2014) on four Nvidia A6000 GPUs with an effective batch size of $512 \times 4 = 2048$ for almost 500k iterations. We utilize a learning rate of $1e{-}5$, a weight decay of $1e{-}4$, $\beta_1 = 0.9$, and $\beta_2 = 0.999$. Following best practices (Goel et al., 2023), we perform augmentations while training. These include randomly rescaling the size and color, flipping, rotating, and translating the center of the body bounding box. To avoid passing in hand crops of extremely low resolutions, we do not rescale the size of the hand bounding boxes. We apply the remaining augmentations on the hands. Across all iterations, we use a weight of 0.05 for 3D keypoint loss, 0.001 on the loss for $\theta_b, \theta_g, \theta_{rh}, \theta_{lh}$, $5e{-}4$ on the loss for $\beta$, $5e{-}4$ for adversarial loss and 0.01 for 2D body keypoint loss. As mentioned in the main text, we gradually increase the loss weights on the 2D hand keypoints. We set the loss weight of the 2D hand keypoints at 0.01 for the first 190k iterations of training. Then, for the next 294k iterations, we boost it to 0.2 to encourage better 2D hand alignment. To process left hands, we adopt the same left-right flipping from Pavlakos et al. (2024).

## A.3 EVALUATION

The Multi-HMR (Baradel* et al., 2024) methods are designed to detect and estimate multiple expressive humans in one-stage. In contrast, all other tested approaches accept crops of each person in an image and individually predict the corresponding human pose parameters. To evaluate the Multi-HMR approaches in the same context as the other methods, we force predictions of individuals. Multi-HMR requires the person's 2D head keypoint to force a predictions on a particular person. In our testing, we estimate this by taking the average of the ground truth ear keypoints. Since AGORA (Patel et al., 2021) and ARCTIC (Fan et al., 2023) provide ground truth ear keypoints for all subjects, we report metrics on the entirety of those validation sets. However, the COCO-Wholebody (Jin et al., 2020; Xu et al., 2022a) validation set contains many instances of occluded ears (the ear keypoints are zeroed out in the ground truth annotations), making it challenging to estimate the head keypoint. For the benefit of the Multi-HMR methods, we prune the COCO validation split to only include subjects with visible ear keypoints.

## A.4 ABLATION

Our final model requires significant GPU resources to train, so we perform the ablation with earlier checkpoints. In particular, we train the single backbone network's head parameters with the same specifics as described in Section A.2 for 190k iterations and employ a comparable version of our model.

| Models | AGORA | | | | ARCTIC | | | | COCO | |
|---|---|---|---|---|---|---|---|---|---|---|
| | MPJPE ↓ | PA-MPJPE↓ | @0.05↑ | @0.1↑ | MPJPE ↓ | PA-MPJPE↓ | @0.05↑ | @0.1↑ | @0.05↑ | @0.1↑ |
| Single Backbone | **141.7** | **60.8** | **0.801** | **0.917** | **101.1** | **46.9** | **0.794** | **0.962** | **0.775** | **0.947** |
| BodhaHMR | 143.3 | 61.0 | 0.797 | 0.916 | 107.6 | 50.8 | 0.775 | 0.947 | 0.744 | 0.923 |

Table 4: **Body results for ablation on hand backbone.**

In Table 4, we report the body results from our ablation study. While the single backbone version consistently outperforms BodhaHMR in this setting, our approach does not lag very far behind. For instance, BodhaHMR is .1% worse on PCK @0.05 for AGORA and 3% worse for COCO. Considering the hand-related improvements we observe in Section 4.4, our method sacrifices minimal body performance to deliver state-of-the-art hand estimations.

