# OpenReview forum: "Reconstructing Humans with Articulated Hands using Transformers"
_ICLR.cc/2026/Conference — ICLR 2026 Conference Withdrawn Submission_

### Official Review · Reviewer_QwTA · 2025-10-31

**Soundness:** 3
**Presentation:** 3
**Contribution:** 3
**Rating:** 4
**Confidence:** 3

**Summary:**

This paper introduces BodhaHMR, a transformer-based method for reconstructing 3D humans with expressive articulated hands from a single RGB image. The approach employs two separate Vision Transformer (ViT-H) backbones—one for body processing (initialized from HMR 2.0) and one for hand processing (initialized from HaMeR). These backbones process crops of the body and hand regions independently, producing feature tokens that are then fused by a unifying transformer decoder to predict SMPL-H parameters. The method is trained on multiple datasets including COCO-Wholebody, Human3.6M 3D WholeBody, and SynthMoCap. Evaluations on AGORA, ARCTIC, and COCO datasets demonstrate state-of-the-art performance on 2D hand pose estimation while maintaining competitive body reconstruction accuracy.

**Strengths:**

1. The method achieves state-of-the-art performance on 2D hand pose estimation across all tested datasets (AGORA, ARCTIC, COCO) while maintaining competitive body reconstruction accuracy, demonstrating the effectiveness of the unified approach.
2. Figures 3-5 demonstrate that the method handles diverse poses, viewpoints, and challenging scenarios including occlusions and egocentric views effectively.
3. The dual-backbone architecture with a coupling transformer decoder is a sensible design choice for balancing coarse body features and fine hand details, addressing a clear gap in existing expressive mesh recovery methods.

**Weaknesses:**

1. The approach primarily combines existing pretrained backbones (HMR 2.0 and HaMeR) with a standard transformer decoder. The main contribution is the coupling strategy rather than novel architectural components, which limits the technical innovation.
2. While 2D hand results are strong, the method underperforms on 3D hand metrics (MPJPE) on ARCTIC compared to the Frankenstein baseline. The paper acknowledges this but doesn't provide sufficient analysis of why the coupling decoder sacrifices 3D accuracy for 2D alignment.
3. Dataset evaluation concerns:
- For COCO evaluation, the validation set is pruned to only include subjects with visible ear keypoints to accommodate Multi-HMR baselines, which may bias results
- No evaluation on egocentric datasets despite showing qualitative egocentric examples

**Questions:**

1. Can you provide more insight into why the method achieves better 2D hand alignment but sometimes worse 3D hand pose compared to Frankenstein? Is this inherent to the coupling approach or a tunable trade-off?
2. How does the method perform when hand crops are of very low resolution or heavily occluded? Are there failure cases that could be characterized?

---

### Official Review · Reviewer_W4kC · 2025-11-01

**Soundness:** 2
**Presentation:** 1
**Contribution:** 1
**Rating:** 2
**Confidence:** 5

**Summary:**

The paper proposes BodhaHMR, a single-image method for expressive human mesh recovery that jointly reconstructs the body and articulated hands using SMPL-H parameters. The system uses two dedicated ViT backbones—one for the full body and one shared for the two hands—whose token outputs are fused by a unifying transformer decoder to regress body pose, hand poses, shape, and camera in a coherent way. Training leverages a mix of in-the-wild, synthetic, and controlled datasets with 2D/3D supervision for bodies and hands; evaluation is conducted on AGORA, ARCTIC, and COCO-WholeBody with both 3D (MPJPE/PA-MPJPE) and 2D (PCK) metrics. Empirically, the method achieves acceptable but not well hand reconstruction performance, while maintaining competitive body accuracy compared to recent baselines.

**Strengths:**

The paper’s main design—separate body/hand encoders with a unifying transformer head—explicitly tackles the cross-part inconsistency that arises when body and hands are predicted independently, yielding a single consistent SMPL-H estimate. This architectural choice is clearly specified and motivated.

**Weaknesses:**

The idea of using separate bounding boxes and encoders/heads for body vs. hands has ample precedent tracing back to early works such as PIXIE and many subsequent studies. If the core contribution is primarily this decomposition plus a fusion head, the architectural novelty feels just incremental.

Wrist–arm consistency not properly evaluated. If the model is initialized from HaMeR and then jointly trained for body+hand, one would expect better wrist–forearm alignment. However, standard MPJPE is largely insensitive to local orientation/kinematic consistency (a wrist joint can have low position error despite poor orientation or continuity). The paper should design targeted evaluations e.g., wrist–forearm relative orientation error, kinematic continuity, or per-frame twist/roll consistency, to validate the intended coupling.

 If a separate hand module or hand-specific crop is used, the paper should compare against hand-focused methods (e.g., WiLoR) rather than primarily full-body baselines. Likewise, beyond ARCTIC, including widely used hand benchmarks such as HO3D and FreiHand would provide a more complete and convincing assessment of hand reconstruction performance.


In Figure 5, the qualitative hand reconstructions (especially rows 3–7) are noticeably weak; in fact, they appear worse than what one would expect from the HaMeR initialization alone. The authors should analyze why full-body joint training degrades hand quality (e.g., loss weighting, or token fusion design ? ) and provide ablations or training diagnostics to identify and explain these not-even-should-happened failure cases.

The paper allocates a relatively large portion of the main text to qualitative figures, some of which do not strongly reinforce the quantitative findings. While visual examples are valuable for demonstrating reconstruction quality, the extensive use of such figures gives the impression that the paper relies on visuals to fill space rather than to provide meaningful analysis.

**Questions:**

I don’t have any major questions — I believe this paper is not yet ready for submission.

---

### Official Review · Reviewer_adMw · 2025-11-02

**Soundness:** 2
**Presentation:** 3
**Contribution:** 2
**Rating:** 4
**Confidence:** 5

**Summary:**

The paper introduces BodhaHMR, a method for reconstructing 3D expressive humans with articulated hands from a single RGB image. Unlike prior approaches that treat body and hand estimation separately, BodhaHMR unifies them to produce consistent and detailed 3D meshes. BodhaHMR is trained on a combination of synthetic, controlled, and in-the-wild datasets: COCO-WholeBody, Human3.6M, SynthBody, SynthHand, using a mix of 2D/3D keypoint, pose, shape, and adversarial losses. BodhaHMR outperforms existing work on AGORA, ARCTIC, COCO dataset.

**Strengths:**

1. Extract the hand separately because of its small resolution makes sense
2. The proposed method outperforms recent baselines.

**Weaknesses:**

1. The results show noticeable misalgnment between the estimated human mesh and the input image
2. No ablation on the design choices and losses, such as fusion scheme, except for backbone.
3. The loss in Eq. 4 seems outdated. Recent methods show better performance with learned priors, like https://dposer.github.io/
4. Not clear why the proposed method outperform existing methods on separated hand and body benchmarks while it is essentially just trained a combined of hand and body datasets.

**Questions:**

Please respond to the concerns in the weakness section.

---

### Note · Authors · 2025-11-13

I have read and agree with the venue's withdrawal policy on behalf of myself and my co-authors.